# Sorting out the Superbugs: Potential of Sortase A Inhibitors among Other Antimicrobial Strategies to Tackle the Problem of Antibiotic Resistance

**DOI:** 10.3390/antibiotics10020164

**Published:** 2021-02-05

**Authors:** Nikita Zrelovs, Viktorija Kurbatska, Zhanna Rudevica, Ainars Leonchiks, Davids Fridmanis

**Affiliations:** Latvian Biomedical Research and Study Centre, Ratsupites 1 k1, LV-1067 Riga, Latvia; nikita.zrelovs@biomed.lu.lv (N.Z.); vkurbatska@gmail.com (V.K.); zhanna.rudevica@biomed.lu.lv (Z.R.); ainleo@biomed.lu.lv (A.L.)

**Keywords:** sortase A, SrtA, sortase A inhibitor, small molecule compounds, antivirulence strategies, antibiotic resistance, superbugs, *Staphylococcus aureus*

## Abstract

Rapid spread of antibiotic resistance throughout the kingdom bacteria is inevitably bringing humanity towards the “post-antibiotic” era. The emergence of so-called “superbugs”—pathogen strains that develop resistance to multiple conventional antibiotics—is urging researchers around the globe to work on the development or perfecting of alternative means of tackling the pathogenic bacteria infections. Although various conceptually different approaches are being considered, each comes with its advantages and drawbacks. While drug-resistant pathogens are undoubtedly represented by both Gram(+) and Gram(−) bacteria, possible target spectrum across the proposed alternative approaches of tackling them is variable. Numerous anti-virulence strategies aimed at reducing the pathogenicity of target bacteria rather than eliminating them are being considered among such alternative approaches. Sortase A (SrtA) is a membrane-associated cysteine protease that catalyzes a cell wall sorting reaction by which surface proteins, including virulence factors, are anchored to the bacterial cell wall of Gram(+) bacteria. Although SrtA inhibition seems perspective among the Gram-positive pathogen-targeted antivirulence strategies, it still remains less popular than other alternatives. A decrease in virulence due to inactivation of SrtA activity has been extensively studied in *Staphylococcus aureus*, but it has also been demonstrated in other Gram(+) species. In this manuscript, results of past studies on the discovery of novel SrtA inhibitory compounds and evaluation of their potency were summarized and commented on. Here, we discussed the rationale behind the inhibition of SrtA, raised some concerns on the comparability of the results from different studies, and touched upon the possible resistance mechanisms as a response to implementation of such therapy in practice. The goal of this article is to encourage further studies of SrtA inhibitory compounds.

## 1. Introduction

The rapid spread of antibiotic resistance throughout the kingdom of bacteria (including, but not limited to, human pathogens that are of healthcare and economic importance) has highlighted the need for alternative means of bacterial disease treatment and reinvigorated the interest in studies that are targeted towards the development of alternative approaches to their containment [1]. Although the search for novel antibiotics and registration of drugs for effective treatment of multidrug-resistant strain infections is likely to continue, the occurrence of resistance due to the natural course of evolution under the selective pressure is inevitable. The question that arises is: Can we keep up the pace of discovery and registration of novel antibiotics [2], with each successive one proving to be effective only for a while, with the rate of antibiotic-resistance emergence which is a complex phenomenon that is still considered incompletely known [3]?

While multidrug-resistant bacterial pathogens are undoubtedly found in both Gram-positive and Gram-negative groups (e.g., two Gram-positive and four Gram-negative highly virulent antibiotic-resistant species are listed in the ESKAPE list [4]), herein we largely focused on the Gram-positive species, as the presence of sortases, the focal point of this paper, is far from being ubiquitous and has been seldom observed in Gram-negative bacterium.

## 2. The Emergence of Antibiotic-Resistant Gram-Positive Pathogens of Healthcare Importance

It was previously noted that numerous major bacterial pathogens that still continue to pose a serious threat to humanity today have emerged in the course of the last 50 years [5]. Some of the most prominent among these are the Gram-positive drug-resistant causative agents of various hospital-acquired (also known as “nosocomial” or “healthcare-associated”) infections. These include (however are not restricted to): The notoriously known multidrug-resistant (MDRSA) and methicillin-resistant (MRSA) *Staphylococcus aureus* strains [6]; *Staphylococcus epidermidis*—for long thought to be a mere opportunistic microorganism, but recently shown to be implicated in medical device-related infections, keratitis, and bacteremia [7]; *Clostridium difficile,* which is considered one of the most frequent causes of hospital-acquired gastrointestinal tract infections [8,9]; different enterococci, some of which (e.g., *Enterococcus faecalis* and *E. faecium*) can be responsible for bacteremia and endocarditis in addition to urinary tract, intra-abdominal, pelvic, and soft tissue infections [10]. Whereas some other (non-nosocomial) relevant Gram-positive pathogens worth a mention are *Streptococcus mutans*—associated with oral diseases and infective endocarditis [11,12]; *Streptococcus pneumoniae*—capable of causing pneumonia, meningitis, sepsis, bacteremia, and otitis [13]; *Listeria monocytogenes*—a significant cause of foodborne listeriosis outbreaks with a fatality rate of up to 30% [14,15]; causative agent of zoonotic anthrax—*Bacillus anthracis*, for which antibiotic resistance may not yet be an urgent matter, but potentially relevant [16,17].

While the antibiotic susceptibility spectra among the different strains of the aforementioned species are highly different and have been described elsewhere (e.g., [7,14,15,17] etc.), all of them have already been documented to harbor resistance to at least one or more antimicrobials in different classes in addition to the high potential for development of further drug resistance [18,19], which signifies the importance of research on the alternative means to combat them.

## 3. Alternative Options

It is, however, a widely known fact that antibiotics are by no means a sole option to treat bacterial pathogens. Although probably the most widely-used and, arguably, the most effective means for treatment of bacterial infections to date, there are also some alternative approaches that were described even prior to seminal discovery of penicillin by Alexander Fleming in 1928 [20]. But, historically, these were slowly dimmed by the wide employment of antibiotics only to resurface recently, and undergo further perfecting along with the various other state-of-art approaches, each with their own advantages and disadvantages in comparison to antibiotics [21].

Further in this article we shall attempt to briefly introduce some of the chosen prospective approaches that are being considered possible alternatives to antibiotics and in the recent years have gathered substantial attention from the research community (Figure 1).

A detailed description of these selected methods, as well as other possible alternatives (e.g., vaccines, antibodies, immune stimulation), is not considered herein because these have been recently summarized and reviewed in detail by other authors elsewhere [1,21,22,23,24,25,26].

### 3.1. Phage Therapy

Bacteriophages (“bacteria eaters”)—viruses of bacteria, the most abundant biological entities in the biosphere—were already being viewed as a tool for treatment of bacterial infections shortly after their independent discovery by Frederick Twort [27] and Felix d’Herelle [28] in the beginning of the 20th century [29]. The usage of strictly lytic bacteriophages in the therapy, although somewhat abandoned during the antibiotic era, has proven to be effective against various human bacterial diseases caused by different cocci, pseudomonads, coliforms, and other pathogens [30]. Some of the benefits of phage therapy, as outlined by Loc–Carrillo and Abedon [31], include: Bactericidal mode of action; “auto-dosing” upon infection of host bacteria population; low inherent toxicity, minimal disruption of normal bacterial flora, narrower potential for inducing resistance, lack of cross-resistance with antibiotics, application versatility, and capability of biofilm clearance. Arguably, the main disadvantage of phage therapy, however, is strongly linked to one of their greatest benefits—the narrow host range of phages, which might span just a few strains of a particular bacterial species. This suggests the need to formulate cocktails of multiple carefully characterized phages with a good proven bacterial killing potential, covering, ideally, different host receptors to rule out the phage-resistance emergence in the course of treatment. However, even then, they are not guaranteed to eliminate every strain of the target species encountered. Despite the fact that strictly lytic phages were almost exclusively used as such natural antibacterial agents for obvious reasons, efforts to use lytic-derivatives of temperate phages with a broad host range to treat bacterial infections in humans were successfully partaken as of recently [32].

### 3.2. Lysins

Lysins—enzymes that are produced by bacteriophages to disrupt the cell wall of their hosts during the last phases of the infection cycle to release the phage progeny into the environment—are also being considered a prospective alternative to antibiotics [33]. Although the ability of lysins to reach the peptidoglycan layer from within the phage-infected cell is usually dependent on another phage-encoded protein (holin) in the natural setting, it has been previously shown that exogenous application of lysins is a rather prospective strategy for treatment of bacterial infections. When applied to Gram-positive bacteria exogenously, lysins have an immediate access to the cell wall, as opposed to being obstructed by the membrane from within the cell, which is the case during the natural phage life cycle; thus, even the small amounts of recombinant phage lysins have demonstrated promising therapeutic potential due to their capability of rapidly lysing the target bacteria. [34]. The first study showing phage lysin effectiveness in vivo was reported in 2001. It described the disease prevention and pathogen elimination in the upper respiratory tracts of mice that were colonized by streptococci [35]. Since then, a plethora of novel lysins have been discovered [36] and numerous advances in the field have been made (recently reviewed in detail by De Maesschalck et al. [37]). The main advantages of lysin therapeutic applications over antibiotics include: Selectiveness regarding their targets, low risk of lysin-resistance emergence, potent activity that occurs within seconds after application of even highly diluted preparations, synergy between different lysins, and lack of toxicity, among others [21]. The main hurdle of “classical” lysin therapy, however, was general ineffectiveness of exogenously applied lysins against Gram-negative bacteria. Nevertheless, this issue is being addressed by the development of state-of-art endolysin-based anti-bacterials termed Artilysin^®^s that are active against Gram-negative pathogens [38].

### 3.3. Antimicrobial Peptides

Antimicrobial peptides (AMPs) are a diverse class of small molecules (generally between 10–50 amino acids long) that are naturally produced by both single-cell and multicellular organisms to either directly kill/inhibit the growth of competing/foreign microorganisms or modulate the innate immune response of the higher organism. These are being considered as yet another alternative tool of combating various pathogens (including antibiotic-resistant bacteria) [39,40]. The main mechanism of action for AMPs lies in their ability to interact with bacterial membranes/cell walls via electrostatic interactions, which results in either rupturing of the membrane or entry into the bacteria with subsequent inhibition of intracellular functions [41]. Although the first antimicrobial peptide lysozyme was discovered by Alexander Fleming as early as 1922 [42], until the 1980s there was a relatively low number of research reports on AMPs [43]. The latest antimicrobial peptide database (APD3) lists 2169 AMPs of various origins, with the vast majority being animal host defense peptides [44].

### 3.4. Bacteriocins

Bacteriocins are highly potent bactericidal agents representing a subclass of antimicrobial peptides produced by bacteria. In general terms these can be classified as either post-translationally modified (class I) or made up of peptides with unmodified amino acids (class II) [45]. The discovery of bacteriocins is attributed to André Gratia, who described the first bacteriocin (colicin V) produced by verotoxin-producing *E. coli* strain to act against other nearby *E. coli* in 1925 [46]. The bacteriocins are structurally diverse and, thus, exhibit different mechanisms of action. Most bacteriocins, however, act by forming pores in bacterial cell membranes, which leads to their disruption and subsequent collapse of the phospholipid bilayer that ultimately leads to death of a bacterial cell [21]. Although their main function is to enhance the competitiveness of bacteriocin-producing bacteria in their natural environment through the elimination of contestant co-inhabitant bacteria, researchers have found applications for bacteriocins in food preservation and clinical setting despite the potential for resistance-development [47,48]. In addition, some of the state-of-art studies have suggested that bacteriocin application might be even more feasible if these peptides are formulated using the state-of-art nanotechnological approaches, and these findings pave way for further enhancements of their usability [49].

### 3.5. Antivirulence Strategies

The rationale behind antivirulence strategies lies in the assumption that “disarmed” pathogens can do no harm; thus, the aim of such strategies is to interfere with the bacterial virulence factors that help the pathogen to either cause damage to the host or evade its immune system and persist within the host [50]. The current anti-virulence strategies tend to target the processes of bacterial quorum sensing systems and biofilm formation ability, as well as to disassemble functional membrane domains and neutralize bacterial toxins via usage of small molecule compounds that show corresponding inhibitory activity against at least one of the virulence factors of the pathogen of interest [51].

It is clearly evident that only a few cells of any pathogenic bacteria are of no immediate concern to the host; however, strength of all pathogens “lies in numbers” that they eventually reach through inevitable propagation in appropriate environmental conditions. With growth of bacterial cell population also grows the necessity for “communication” between the cells, which is essential for coordination of the whole community towards situations that are advantageous to a population as a whole. This communication was revealed to be mediated by small signal molecules that auto-induce expression of particular genes [52]. The cell density-dependent process of such communication is termed “quorum sensing” [53] and there are indications that it might occur even between different species [54]. It is thought that as much as 4–10% of bacterial genome and ≥20% of proteome could be influenced by quorum sensing, and the effects of quorum sensing range from harmless metabolic or phenotypic adaptations to modulation of pathogenicity related virulence (e.g., biofilm formation, toxin expression) [55]. Thus, interference with quorum sensing cascades that blocks bacterial virulence-associated “communication” (“quorum quenching”) was outlined as a promising antivirulence strategy [56].

Whereas inhibition of the transcription might seem a more appropriate approach for some of the bacterial virulence factors, such as protein-based toxins, thus, eliminating the cause rather than a consequence, it is also possible to target these molecules after their synthesis. Studies in this direction have led to the development of approaches that enable evasion from the destructive effects of toxins on the host through their neutralization by antibodies, toxin activity blocking by small molecule compounds [57], or even sequestration of toxins in artificial liposomes [58].

Recently discovered bacterial functional membrane microdomains (FMMs), which resemble lipid rafts of eukaryotic cells in both structure and function, were immediately proposed as yet another novel antivirulence strategy target due to the involvement of FMMs/FMM-associated proteins (e.g., flotillins) in biofilm formation, attachment, virulence, and signaling [59,60,61].

The ability of some pathogens to form biofilms, which are bacterial communities enclosed in structure formed by thereof extracellular matrix components, helps embedded bacterial communities to evade host defensive responses, mitigate environmental stresses, and provide protection from antibiotics [62]. Pathogenic biofilm formation on the host tissues facilitates the onset of a chronic bacterial disease that is considerably more difficult to treat than acute infections caused by pathogens in the planctonic (“free”) state [63]. Thus, the prevention of biofilm formation is also recognized as a prospective antivirulence strategy that strives to limit bacterial adhesion to surfaces or affect the extracellular matrix component production, and sometimes even destroy the extracellular matrix post factum, when biofilm has already been formed [51].

One of the particularly promising antivirulence strategies that is aimed at Gram-positive bacteria is the inhibition of bacterial cysteine protease—sortase (especially SrtA)—activity by small molecule compounds [64].

## 4. Sortase A (SrtA)

Sortases are cysteine transpeptidases that play a pivotal role in shaping the architecture of microorganisms by mediating the covalent protein attachment to their cell wall [65]. While sortase enzymes are found ubiquitously in Gram-positive bacteria, where their functions have been largely elucidated, genes encoding proteins belonging to the sortase superfamily have also been documented in a fraction of genomes from Gram-negative and archaeal species, although their functions in these organisms have not yet been completely understood [66]. Despite the fact that up to eight different classes of sortases (A, B, C, D1, D2, E, F, and “Marine”) are already being recognized as of now, the “canonical” *Staphylococcus aureus* SrtA still remains the most extensively studied enzyme of this group [66,67,68,69,70].

SrtA is a membrane-associated cysteine protease that catalyzes a cell wall sorting reaction by which surface proteins, including virulence factors, are anchored to the bacterial cell wall. The steps involved in the SrtA-assisted protein anchoring to the cell wall of Gram-positive bacteria have been uncovered and, in the case of *Staphylococcus aureus*, are as follows [71]: (1) Proteins with the N-terminal secretion signal peptide are transported to the cell surface via the secretory (Sec) pathway; proteins that are destined to be anchored to the cell-wall contain an additional LPXTG (leucine–proline–any residue–threonine–glycine) motif followed by a hydrophobic region and a tail of charged residues within their C-terminus; (2) the LPXTG motif is recognized by the membrane-associated SrtA enzyme; (3) cleavage between T and G of a LPXTG motif is introduced by two-step transpeptidation reaction and threonine is covalently attached to the cell wall amino group of a pentaglycine [72].

It has previously been shown that the C-terminal sorting signal with a highly-conserved LPXTG motif is prevalent within cell wall-anchored surface proteins of Gram-positive bacteria [73] and the elucidation of Gram-positive bacteria surface protein roles in interactions with the host, which also includes virulence, is a topic that has gathered substantial research attention in the past. These studies have led to hypotheses that it might be possible to reduce pathogen virulence by interfering with the display of the aforementioned proteins on the surface of the cells [74,75,76]. However, it was not before the seminal study of Mazmanian and colleagues [77], which demonstrated that the SrtA enzyme is an absolute necessity for surface protein anchoring to the cell wall envelope and consequent pathogenesis of *S. aureus* infections, that the sortase inhibitor research era really begun. The primary goal of the subsequent studies was to find a way to disrupt the pathogenesis of bacteria without affecting microbial viability, thus, treating infections caused by Gram-positive pathogens. In addition to *S. aureus*, a decrease in virulence due to inactivation of SrtA activity has also been demonstrated in *Listeria monocytogenes* [78], *Streptococcus pneumoniae* [79], *S. suis* [80], and *S. mutans* [81].

## 5. SrtA Inhibitors

In comparison to the conventional antibiotics and other anti-virulence strategies, the main advantage of SrtA inhibitor usage as a potential Gram-positive pathogen infection treatment lies in the relative harmlessness of possible resistance emergence to the host [82]. We hypothesize that, if the inhibitor is properly designed and targeted at the substrate binding site, the most likely outcome of the inhibitor-induced selective pressure shall be either in the formation of a mutated SrtA gene that produces enzyme with an altered inhibitor binding site, which most probably shall also alter the enzymatic core structure, decreasing enzyme activity and, along with it, overall virulence of the pathogen, or increase the production of sortases to cope with the decrease in enzymatic activity, which shall result in an increased metabolic burden and decreased pathogen proliferation rate. In either way, acquisition of resistance shall come with the cost that decreases the overall pathogenicity of the target bacteria. Alternatively, pathogens might also start to produce a novel protein that either enzymatically alters the inhibitor, thus abolishing its inhibitory properties, or binds it to effectively remove the inhibitor from the surrounding environment. However, we believe that such a scenario is unlikely, as it would require the coincidence of far too many favorable factors. This, in our opinion, provides a prospective of the fail-safer therapy concept ultimately resulting in the greater success rate regardless of the resistance emergence—the pathogen is either being “disarmed” by the designated SrtA inhibitor treatment or is “significantly hampered” in the course of evolution under the selective pressure imposed by such treatment. This situation, however, would differ significantly from the previously described if the selected inhibitor were to work as an allosteric modulator that could bind to the enzyme at other sites than the active center. In this case it is highly probable that the exerted selective pressure shall result in a development of metabolically or enzymatically unburdened resistance through the introduction of only the allosteric binding site altering mutation that would not change the overall molecular structure of the enzyme molecule. Therefore, we believe that the development of sortase inhibitors should be exclusively directed towards the identification of molecules binding to the enzymatic core. An alternative approach for treatment, of course, might also be an employment of cocktails of sortase inhibitors that couple to different allosteric binding sites, as it is highly unlikely that several inhibitory effect inactivating mutations would occur at the same time. However, such an approach would require a careful characterization of each individual compound and their interactions with the sortase, as well as elucidation of possible side effects on the health of the host, prior to being employed for the treatment of bacterial infections in a therapeutical setting, while implementation of such extensive thorough studies undoubtedly requires lengthy allocation of significant financial and labor resources.

Conceptually different strategies for the initial discovery of novel potential SrtA inhibitors have previously been adopted in the field, including: Screening of natural products [83], small compound library high throughput screening [84], virtual screening in silico [85], and fragment based lead discovery [86]. Although initial hits can be generated by each of the approaches individually, these approaches complement each other well, and, thus, combining them in a single study could yield better results. While working on this article and exploring the repertoire of uncovered sortase inhibitor leads (extensively summarized in Appendix A [84,87,88,89,90,91,92,93,94,95,96,97,98,99,100,101,102,103,104,105,106,107,108,109,110,111,112,113,114,115,116,117,118]), we observed that despite the great advances in this field, there was also a great ambiguity in the acquired results.

The presentation of the determined compound-specific parameters differed significantly as IC_50_ values were presented both as molar and mass concentrations, with the latter being less informative due to the great differences in tested compound molecular weight. Even more, the minimum inhibitory concentration (MIC) test has not been performed for many of the reported leads; thus, their bacterial toxicity and possible drawbacks of their applicability in anti-virulence therapies remains unexplored. An additional aspect that seems to hamper the effective cross-study comparison of various compounds is related to the basic enzyme kinetics, because, as clearly demonstrated by the data that we acquired in one of our latest studies (Appendix A and Figure 2) where, while searching for suitable substrate, we tested the same SrtA-compound combination with two different substrates, and acquired IC_50_ values were dependent on a combination of both the selected substrate and tested compound. Thus, this parameter is usable for compound comparison only if the overall setting of the experiments is identical. This situation clearly demonstrates that standardization of experimental procedures and results’ presentation is needed in this field to improve cross-study comparison capabilities and achieve a greater whole SrtA research community success rate.

It is also worth noting that only few of the studies on the identification of SrtA inhibitory compounds that were reviewed in this article have actually taken their studies further and performed the actual efficacy evaluation for their most potent compounds beyond mere documentation of their inhibitory effects in enzymatic models (e.g., no MIC tests or in vivo models), which suggests that there is a need for follow-up in-depth studies of these compounds to evaluate the plausibility of their development into an actual therapeutic agents of a new kind. This necessity is further highlighted by the fact that, to the best of our knowledge, none of the thus far identified SrtA inhibitors has yet been advanced to the clinical trials, while numerous products from some other classes of possible antibiotic alternatives (e.g., phage therapy, lysins) are already seeking medical approval by moving through the different phases of clinical trials.

Therefore, we believe that, in the near future, assessment of the previously described SrtA inhibitor efficacy should be conducted in the natural setting using a standardized approach, with the acquired results made available to the scientific community as soon as possible, so that aggregation of the pro and cons evidence would either enable the consolidation of the SrtA inhibitor position among the potential antibiotic alternatives, thus drawing more attention from the funding bodies and attracting additional researchers to the field, or prove the inefficiency of this approach, thus enabling the redirection of resources to the further development of approaches that would prove to be more reasonable.

## 6. Materials and Methods

### 6.1. SrtA Expression and Purification

The presumed catalytic core of the SrtA gene from *Staphylococcus epidermidis* and *Streptococcus pneumoniae* microorganisms that encode enzymatically active transpeptidase domains were PCR amplified and inserted in the vector for bacterial expression [119,120]. The DNA sequence encoding for N-terminally truncated *Staphylococcus epidermidis* SrtA (Se-SrtA∆N70) was PCR-amplified from genomic DNA of *S. epidermidis* YC-1 strain using 5′-TATACATATGGGTTATATAGAAGTTCCAGATG-3′ and 5′-TAATCTCGAGTTAGTTAATTTGTGTAGCTATG-3′ primers. The DNA sequence encoding for N-terminally truncated *Streptococcus pneumoniae* SrtA (Sp-SrtA∆N59) was PCR-amplified from genomic DNA of the *S. pneumoniae* D39 strain using 5′-ATTACATATGGAAGAAAATCAGGATACAGAAG-3′ and 5′-TATACTCGAGTTAATAAAATTGTTTATATGGT-3′ primers. The PCR-amplified DNA sequences were cloned into the pET28b (Novagen, Madison, WI, USA) vector through NdeI and XhoI restriction sites for expression as His-tagged proteins in *E. coli*. The plasmid DNAs were isolated using the Plasmid Miniprep Kit (Thermo Fisher Scientific, Waltham, MA, USA), verified by restriction analysis, and approved by sequencing using the BigDye™ Terminator v3.1 Cycle Sequencing Kit (Thermo Fisher Scientific, Waltham, MA, USA).

The recombinant sortases were expressed in *E. coli* BL21 (DE3) cells using a standard protocol and purified as previously described [121]. In short, transformed bacteria were cultured in Luria–Bertani medium at 37 °C until an optic density at 600 nm (OD_600_) of 0.6–0.7 was reached. Recombinant protein synthesis was induced at 37 °C, 200 rpm by adding isopropylthiogalactoside to a final concentration of 1 mM and continued for 3 h. The cells were then sedimented by centrifugation and resuspended in phosphate-buffered saline buffer pH 7.5 with 1 mM dithiothreitol, 0.1% Triton X100, and protease inhibitor cocktail added prior to sonication. The recombinant soluble sortases were further purified under native conditions.

Both histidine-tagged recombinant SrtA proteins were purified by affinity chromatography using the Ni-nitrilotriacetic acid agarose (Qiagen, Hilden, Germany) pre-equilibrated with buffer A (10 mM imidazole, 1 mM DTT, 50 mM NaH_2_PO_4_, and 300 mM NaCl, pH 8.0). Protein binding was performed for 1 h at room temperature on the rotator and 20 mM imidazole buffer A pH 8.0 was then used during the wash procedure to remove unbound proteins. The proteins were eluted by adding 300 mM imidazole buffer A to the column and then additionally purified on Superdex75 120 mL packed on XK16/70 gel filtration column (GE Healthcare, Chicago, IL, USA) by using 1 mM DTT, 50 mM Tris-HCl, and 150 mM NaCl pH 7.5 end buffer. Purified proteins were concentrated to 10 mg/mL by centrifugation at 4000 rpm at 4 °C in Amicon Ultra centrifugal filter units with a 3 kDa cut off (Millipore, Burlington, VT, USA). Further clarification of protein was achieved by centrifugation at 16,400 rpm for 10 min at 4 °C, and 20% glycerol was added to the buffer for prolonged protein storage on ice. 

### 6.2. FRET Enzymatic Assay and IC_50_ Determination

The tested compounds were commercial products purchased from Enamine Ltd. The half maximal inhibitory concentration (IC_50_) values for compounds were determined by monitoring the increase in fluorescence intensity upon cleavage of the Dabcyl-LPETG-Edans FRET peptide (excitation/emission wavelength of 360/485 nm) or Abz-LPETG-K (Dnp) FRET peptide (Ex/Em 320/420 nm), which were used as the substrates for both sortases [108]. In brief, the test compounds of various concentrations (3–100 μM) were added to 1–20 μM of each of the recombinant sortase in 20 mM HEPES, 5 mM CaCl2, 0.05% Tween-20 buffer pH 7.5 [122]. Subsequently, the peptide Dabcyl-LPETG-Edans at a final concentration 16–32 μM (depending on each sortase enzymatic activity) was added to the reaction. Similarly, test compounds were added to the 1–20 μM recombinant sortases with subsequent peptide Abz-LPETG-K (Dnp)-NH2 addition at a final concentration 1 or 5 μM (depending on each sortase enzymatic activity). Fluorescence was recorded for 15 h within an interval of 30 min at 37 °C temperature using a Tecan F200 microplate reader. IC_50_ values were calculated using GraphPad Prism Software [123].

## 7. Conclusions

To conclude, while SrtA activity inhibition by small molecule compounds that leads to a decrease of Gram-positive pathogen virulence seems to be a reasonable approach to combat drug-resistant pathogens and many of such compounds are displaying promising potency, the SrtA research community still has a long way to go before a definite conclusion on the feasibility of this strategy can be reached. We believe that employment of standardized procedures (both in vitro and in vivo, with an employment of appropriate controls to elucidate the non-specific effects) for testing of identified leads and reporting of the acquired results in a standardized manner that would rapid cross study comparison (MIC tests, k_i_ value calculations, etc.) should be a top priority of the field. Collectively defining the recommended workflows and/or guidelines is absolutely possible and would prove to be extremely beneficial for a relatively small SrtA research community, possibly leading to a faster accumulation of knowledge in the field while ensuring comparability of the results.

## Figures and Tables

**Figure 1 antibiotics-10-00164-f001:**
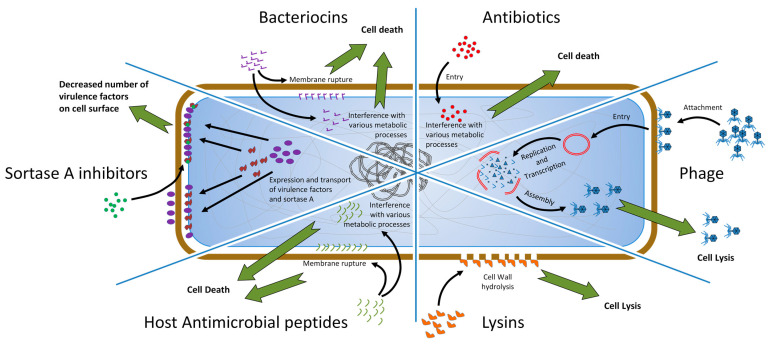
Conceptual sexpartite diagram comparing simplified modes of action for antibiotics and some of the alternative approaches to combat the pathogenic bacteria. The diagram is divided into six parts, each corresponding to the given approach for combating the Gram-positive pathogenic bacteria. Brown rounded rectangle represents peptidoglycan layer and together with the contents represents the Gram-positive bacterial cell. Perimeter of the blue rounded rectangle represents the plasma membrane. Black arrows represent interactions and point at the desired spatial location/locations of the agents for interplay with their bacterial targets. Green arrows pointing away from the cell represent the desired outcome of an interplay between a given agent and its target. Colored figures unique to each of the diagram parts represent agents that can be viewed as an alternative to antibiotics (for example red circles represent molecules of antibiotics). In the case of sortase A (SrtA) inhibitors (leftmost part of the diagram): Green circles—inhibitory compounds, purple ellipses—virulence factors, amorphous brown figures—SrtA enzyme.

**Figure 2 antibiotics-10-00164-f002:**
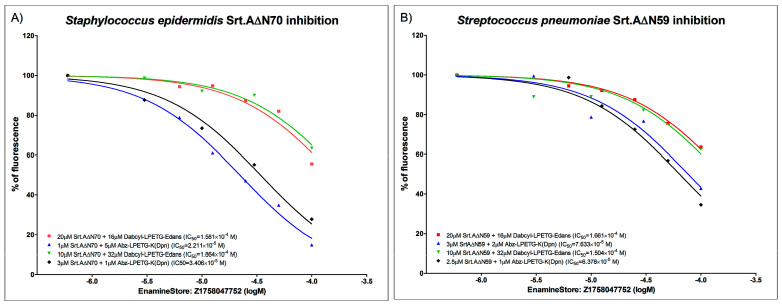
Enzyme inhibition curves for (**A**) recombinant Δ70 *Staphylococcus epidermidis* SrtA, and (**B**) recombinant Δ59 *Streptococcus pneumoniae* SrtA. N-terminal parts containing membrane anchor domain were cleaved to improve solubility and purification of enzyme. FRET-based inhibition assays were carried out employing two different substrates: Abz-LPETG-K (Dnp), Dabcyl-LPETG-Edans, one lead compound, Z1758047752, which was acquired from Enamine Ltd. (https://www.enaminestore.com/). Measurements were performed in triplicates and repeated twice. Results demonstrate that acquired IC_50_ values are substrate dependent.

## Data Availability

The data presented in this study are available in Appendix A.

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
