# Peer review of "Sorting out the Superbugs: Potential of Sortase A Inhibitors among Other Antimicrobial Strategies to Tackle the Problem of Antibiotic Resistance"

_antibiotics, 2021, doi:10.3390/antibiotics10020164_

Round 1
Reviewer 1 Report
The authors prepared a manuscript in which they were supposed to talk about new Sortase A inhibitors.
However, the layout of the manuscript is unclear.
The abstract does not represent the manuscript correctly. The material and methods part has only two bibliographic entries and should be described in more detail.
The number of references to experimental work are too exaggerated. Some data in table S1 do not correspond to those presented in the cited paper. Have they been retested in this esxperiment?
What is the purpose under Chapter 3 in the discussion on inhibitors Sortase? In my opinion only 3.5 should remain. Where is table A1 located?
Bibliographic entry 50 is not compatible with what is described in the manuscript.
Author Response
Comments and Suggestions for Authors
The authors prepared a manuscript in which they were supposed to talk about new Sortase A inhibitors.
However, the layout of the manuscript is unclear.
Response: The aim of this paper is stated in manuscript’s cover letter and it also explains the contents of the manuscript.
We agree that the layout of the manuscript might seem inappropriate for article that focuses on Sortase A inhibitors, however, it was planned and intentionally written in such way, as the covered topics are broader than perceived by the respected Reviewer 1. This particular “perspective” (Sorting out the superbugs: potential of Sortase A inhibitors among other antimicrobial strategies to tackle the problem of antibiotic resistance) was prepared for submission to the special issue "Solutions to Antimicrobial Resistance", which, to our understanding, covers a rather broad and diverse topic that, we believe, shall appeal to both, more general readership and investigators from specific topic-related areas, alike. We also agree with the Reviewer 1 that from a first glance the layout might feel redundant and, arguably, less relevant to more experienced investigators from sortase A research community. However, we believe that providing a rather concise introduction to some of the antibiotic resistant pathogens that are of concern to medical specialists and briefly mentioning, in our opinion, most prominent alternative strategies for tackling them is necessary to explain the problem and possible solutions in a broader context before narrowing down to a very specific and promising area of investigation (SrtA inhibition studies), that is, arguably, one of the globally least known antivirulence strategies. In our opinion, this serves two main purposes: 1) the layout helps to provide readers with some context for better understanding of SrtA inhibitor place among other possible AMR solutions, which, we think, complies with the topic of this special edition; 2) inclusion of a somewhat “holistic” approach to the topic would improve the discoverability of the article, should it be accepted for publication, and introduce more general readership to a selected area of investigation (SrtA inhibition), thus popularizing it.
In our opinion, the layout of this “perspective” format manuscript is also justified by the fact that it might appeal to a wider readership, because it does not hinder the discoverability of SrtA inhibitor-related contents of the manuscript for investigators from this particular field, while making sure that readers from other fields are not repelled by an overwhelming field specific complexity. Thus, we feel that the layout of the manuscript shall help to “raise more general interest in alternative strategies of combating the drug-resistant pathogens, while also motivating much needed discussions in the SrtA research community”, which is intended aim of the manuscript, as stated in the cover letter.
The abstract does not represent the manuscript correctly.
Response: In regards to the abstract, unfortunately, it is not clear to us what causes the Reviewer’s 1 concern. We have tried to write the abstract with a “background/specific topic/discussion” structure that, in our opinion, fits the perspective type of manuscripts and reflects the contents of the full text. In the abstract (which now contains minor revisions to improve the language quality) we have started with the problem and rationale statement, which explains the global need of alternative approaches for treatment of antibiotic resistant pathogen infections (lines 8-16), which is followed by an introduction of the selected topic in question (SrtA and its inhibition; lines16-20) and concluded with the summary of specific points that are being discussed from the authors viewpoint (lines 20-25). Although we believe that our abstract is an accurate representation of the manuscript, we would be happy to receive more specific suggestions for its improvement.
The material and methods part has only two bibliographic entries and should be described in more detail.
Response: We think that methodology is described in sufficient detail (especially for this “perspective” type of manuscript where the data acquired by the authors is by far not the main thing that is being discussed). Although we believe that the description allows replication of the results, if the Reviewer suggests that elaboration on something particular about the methodology is necessary, we would be happy to include this in the next round of revision. At this point more experimental papers were referenced to in the section and now the section includes 6 bibliographic entries.
The number of references to experimental work are too exaggerated.
Response: There are 33 references (bibliographic entries 87-119, ~30% of total) that correspond directly to the experimental work on the basis of which our manuscript was written (these are also summarized in table S1). While writing this manuscript, we have strived not only to give our personal viewpoint on the subject, but also to make sure that the experimental background is referenced and covered in a complete manner, thus allowing the readers to acquaint themselves with the most prominent studies of the field. We believe that the summary of experimental work that was conducted by researchers worldwide throughout the years serves as an evidence to the urgency of the topics discussed and issues raised. Since side-by-side comparison of these studies/compounds is one of the focal points of this manuscript, we find it incredibly hard to exclude any of these experimental studies as it would involve complete exclusion of affiliated SrtA inhibitory compounds and would lead to a biased approach to the summary presented.
Some data in table S1 do not correspond to those presented in the cited paper. Have they been retested in this esxperiment?
Response: We thank Reviewer 1 for this question. The data on the 3-[3-(Tetrahydro-3-thiophenylamino)-1-piperidinyl]-2(1H)-pyrazinone SrtA inhibition was included in the table S1 on the basis of experimental work described in our manuscript (see Materials and Methods section; Fig. 2).
Apart from that, within the scope of our experiments, we did not re-test the potency of any of the compounds seen in table S1, and the data seen in the table are taken or, if necessary, derived from the study referenced in the “Reference” column of the table. However while writing the manuscript we did notice that some of the referenced studies might have re-tested potency of the other compounds that are mentioned in the table S1, therefore some of the values might slightly differ for a given compound from those that were acquired in other studies. However, we have tried to trace back to the original study that has shown and measured the SrtA inhibitory effect for any identified compound, which was then chosen as a reference for the values seen in table S1.
The authors have re-checked the table S1. Reviewer 1 is correct, a wrong reference was provided for Chlorogenic acid (15th compound from the top of the Supplementary table 1), this has now been corrected. MW for some of the compounds were further refined to include more digits after the decimal point for calculations, this has minorly (in the scope of second digit after the decimal) changed the derived IC50/MIC values that were not given in the referenced paper. We have now also renamed some of the substrates (same substrates referenced to differently by different authors) to ensure consistency in naming.”Type” column values have now seen some additions that were missed during the table generation. Column previously called “In vivo” now is renamed to “in vivo studies carried out” to ensure disambiguation of possible misinterpretation.
Some of the cited papers might not have indicated the molecular weight of the compounds, which was then derived from other sources (e.g., calculated from molecular formula or extracted from PubChem). Others have included a previous reference instead of an exact data (e.g. SrtA used for IC50 determination), which was followed to acquire the data. Also, for compound potency overview purposes MIC and IC50 values were often converted from μM to μg/mL or vice-versa if one of these values were not presented.
We have now meticulously reviewed and compared the data presented in the table S1 with the cited sources and failed to identify any other issues, we would be very thankful if Reviewer 1 could be more specific about the issues that were identified during review process. We are eager to correct any remaining technical error or incorrect interpretation, should it still be present in the second version of the table S1. We understand that this is a potentially very serious issue and we would appreciate if the respected Reviewer 1 would elaborate on it during the second round of manuscript revision.
What is the purpose under Chapter 3 in the discussion on inhibitors Sortase? In my opinion only 3.5 should remain.
Response: The purpose of “Chapter 3” has already been elaborated in the earlier answer, where we tried to explain the rationale behind the layout of the manuscript (as seen in the first version of the manuscript). However, if the Reviewer 1 insists that inclusion of the majority of “Chapter 3” in the manuscript is not justified enough, and the manuscript would arguably benefit from its exclusion (only so that subsection 3.5. remains), it can be removed/shortened in further round of revisions.
Where is table A1 located?
Response: We are extremely sorry for this mistake and see why Reviewer 1 found it puzzling. The table A1 and S1 are the same supplementary table, which was referenced inconsistently throughout the text. This has now been addressed and erroneous occurrence of “table A1” in line 266 has now been changed to “table S1”.
Bibliographic entry 50 is not compatible with what is described in the manuscript.
Response: (Reference and line numbering as seen in 1st version of the manuscript)
Bibliographic entry 50 was erroneously referenced twice in the original version of the manuscript. The correct usage of reference to bibliographic entry 50 is in line 181 (subsection 3.5. “Antivirulence strategies”). The usage of bibliographic entry 50 in the line 333 was an error (subsection 6.1. “SrtA expression and purification”). The correct reference for line 333 has been inserted. We are sorry for this typo and this issue has now been corrected in the second version of the manuscript.
The authors appreciate the efforts partaken by the respected Reviewer 1 in helping us to improve this manuscript and fix the errors identified. We have tried to explain our point of view regarding the commentaries provided and address the issues found in the manuscript/supplementary table 1. If our answers and implemented changes do not convince the Reviewer 1, we will be ready to address any remaining issue in the following round of revisions.
Reviewer 2 Report
The authors tackle an important topic, which is that of antibiotic resistance, and consider an approach that seems extremely promising. Please find below some specific corrections and some more global considerations
Abstract
Line 14-16: please check the sentence
Introduction
Line 40: Gam instead of Gram
Line 100-101: please check the sentence
Line 125: please check the sentence
Line 142: please check Artilysin®s
Lines 228-243: I really hope the onset of resistance to Srt A inhibitors is as rare and unlikely as the authors claim, but it is not clear on the basis of which data and evidence the authors make such claims. Therefore, I urge them to provide evidence to support their theory
Fig.2: please add IC50 for every compound and condition to facilitate reading the graph (namely in graph 2B it is difficult to identify the curves)
Line 299, 303, 307, 365, 368, 375: srtA is in lowercase
Lines 266 and 278: are table A1 and table S1 the same? In supplementary material I only found one table.
Lines 303-310 could be moved to it could be moved and complement the conclusion, which reiterates more or less the same concept.
Many double spaces throughout the manuscript
Overall the manuscript is interesting and covers a very important aspect concerning the search for new approaches to solve the problem of antibiotic resistance. the authors have carried out an accurate review of the present literature and perhaps, precisely for this reason, I find it hard to consider the manuscript a "perspective" rather than a review, especially in light of the fact that the topic, as pointed out by the authors themselves, is already the subject of studies by numerous groups that the article itself documents in detail in the supplementary table.
Therefore, I ask the Editor and authors to carefully evaluate whether the definition "perspective" is adequate for this manuscript. I also point out that some sentences, namely in the introduction (lines 50-65 is one single sentence!), are verbose and really difficult to read, therefore I ask the authors to lighten them and perhaps simplify them to facilitate reading.
Author Response
Comments and Suggestions for Authors
The authors tackle an important topic, which is that of antibiotic resistance, and consider an approach that seems extremely promising. Please find below some specific corrections and some more global considerations
Abstract
Line 14-16: please check the sentence
Response: The sentence was rewritten and the last sentence of the abstract seen in the original version of the manuscript (lines 23-24: “The goal of this article is to encourage further studies of SrtA inhibitory compounds.”) has now been removed to avoid abstract word count restriction violation.
Following changes were introduced:
Original manuscript: “While drug-resistant pathogens are undoubtedly represented by both Gram(+) and Gram(-) bacteria, sortase A (SrtA) inhibition seems perspective among the at Gram-positive pathogen targeted antivirulence strategies, but it still remains less popular than other alternatives.”
Second version of the manuscript: While drug-resistant pathogens are undoubtedly represented by both Gram(+) and Gram(-) bacteria, possible target spectrum across the proposed alternative approaches of tackling them is variable. Although sortase A (SrtA) inhibition seems perspective among the at Gram-positive pathogen targeted antivirulence strategies, it still remains less popular than other alternatives.
Introduction
Line 40: Gam instead of Gram
Response: In the second version of the manuscript this typo has been corrected.
Line 100-101: please check the sentence
Response: In the second version of the manuscript the sentence was rewritten in a following manner:
Original manuscript: “Bacteriophages (“bacteria eaters”) – viruses of bacteria, the most abundant biological entities in the biosphere and, as natural predators of bacteria, were already considered as a method for treatment of bacterial infections shortly after their independent discovery by Frederick Twort [27] and Felix d’Herelle [28] in the beginning of the 20th century [29].”
Second version of the manuscript:
“Bacteriophages (“bacteria eaters”) – viruses of bacteria, the most abundant biological entities in the biosphere were already being viewed as a tool for treatment of bacterial infections shortly after their independent discovery by Frederick Twort [27] and Felix d’Herelle [28] in the beginning of the 20th century [29].”
Line 125: please check the sentence
Response: The sentence within Lines 124-131 of the original manuscript was rewritten in a following manner to facilitate the readability of this paragraph:
Original manuscript: “Although they usually are dependent on an another phage-encoded protein (holin) to be able to reach the peptidoglycan layer from within the cell, it has been previously shown that, when applied to Gram-positive bacteria exogenously, even small amounts of recombinant phage lysins show high therapeutic potential and are capable of rapidly lysing the bacteria, because in such case, lysins have an immediate access to the cell wall, as opposed to being obstructed by the membrane from within, which is the case of natural phage life cycle [34].”
Second version of the manuscript: “Although the ability of lysins to reach the peptidoglycan layer from within the phage-infected cell is usually dependent on another phage-encoded protein (holin) in the natural setting, it has been previously shown that exogenous application of lysins is a rather prospective strategy for treatment of bacterial infections. When applied to Gram-positive bacteria exogenously, lysins have an immediate access to the cell wall, as opposed to being obstructed by the membrane from within the cell, which is the case during the natural phage life cycle, thus, even the small amounts of recombinant phage lysins have demonstrated promising therapeutic potential due to their capability of rapidly lysing the target bacteria. [34].”
Line 142: please check Artilysin®s
Response: Artilysin is a registered trademark in the European Union, United States, and some other countries. The spelling is correct, however, registered trademark symbol symbol (®) has now been superscripted for the correct registered trademark symbol formatting purposes.
Lines 228-243: I really hope the onset of resistance to Srt A inhibitors is as rare and unlikely as the authors claim, but it is not clear on the basis of which data and evidence the authors make such claims. Therefore, I urge them to provide evidence to support their theory
Response: Lines 228-243 of our original submission are dedicated to discussion of various resistance emergence scenarios and, in some cases, possible likelihood of their occurrence in comparison to other scenarios. None of these discussions are touching the subject of overall probability of resistance occurrence. Even more, we believe that, at the moment it is not possible to obtain the data on rarity and unlikeliness of SrtA inhibitor resistance emergence, because, to our best knowledge, there are no approved drugs that would target SrtA and inhibition of this enzyme is not a widely practiced approach, thus there are far too few opportunities for the emergence of resistance to draw any statistically reliable conclusions without specifically conducting such studies. Also, we strongly believe that such resistance shall emerge, but it will come with the cost that shall be related to moderate decrease of pathogen virulence, as it is often in the case of antibiotic resistance mutation formation that are accompanied by the decrease in overall fitness (PMID: 25861385). To emphasize that these lines should be considered as a theoretical speculation, we have augmented the second sentence of the indicated paragraph with “We hypothesize that…”. We believe that this section is important to facilitate the discussions on the subject within the SrtA community and could possibly stimulate studies on the target bacteria evolution under the selective pressure imposed by SrtA inhibitory compounds.
Fig.2: please add IC50 for every compound and condition to facilitate reading the graph (namely in graph 2B it is difficult to identify the curves)
Response: We thank Reviewer 2 for this valuable suggestion, IC50 values for every curve were added to the corresponding figure legends in the second version of the figure, the curves are now also colored differently to facilitate their identification.
Line 299, 303, 307, 365, 368, 375: srtA is in lowercase
Response: Occurrences of “srtA” in lines mentioned by Reviewer 2 have now been changed to “SrtA” to ensure consistent formatting throughout the text.
Lines 266 and 278: are table A1 and table S1 the same? In supplementary material I only found one table.
Response: We are sorry for this inconsistency in naming of supplementary table. As correctly pointed by Reviewer 2, the table A1 and S1 are, indeed, the same supplementary table. This has now been addressed and erroneous occurrence of “table A1” in line 266 references has now been changed to “table S1”.
Lines 303-310 could be moved to it could be moved and complement the conclusion, which reiterates more or less the same concept.
Response: While we thank the Reviewer for this suggestion, and we believe that in “perspective” format manuscripts conclusions are somewhat interchangeable with parts of the discussion and might even be presented “seamlessly” in the discussion, we have tried to include a minor dedicated “conclusion” section, which we think, would suffer from being made longer. Although this can be done in the further round of revision, we think that it then might actually be better to exclude the dedicated conclusion section and append the contents as the last paragraph of the discussion.
Many double spaces throughout the manuscript
Response: We are sorry for this formatting mistake that was initially overlooked by us. We, indeed, identified many occurrences of double spaces (>10). These now have been changed to a single space.
Overall the manuscript is interesting and covers a very important aspect concerning the search for new approaches to solve the problem of antibiotic resistance. the authors have carried out an accurate review of the present literature and perhaps, precisely for this reason, I find it hard to consider the manuscript a "perspective" rather than a review, especially in light of the fact that the topic, as pointed out by the authors themselves, is already the subject of studies by numerous groups that the article itself documents in detail in the supplementary table.
Therefore, I ask the Editor and authors to carefully evaluate whether the definition "perspective" is adequate for this manuscript. I also point out that some sentences, namely in the introduction (lines 50-65 is one single sentence!), are verbose and really difficult to read, therefore I ask the authors to lighten them and perhaps simplify them to facilitate reading.
Response: We are flattered by these Reviewer 2 commentaries, as in this manuscript we have, indeed, attempted to include an accurate (although a rather concise) review of the relevant literature on the subject of SrtA inhibitors. However, we have found that the “perspective” format was the best fit for the contents of the manuscript, because after briefly giving readers some broader context and topic-specific evidence that is necessary to evaluate the importance of issues being raised and back up the points that are made, the manuscript largely involves presentation of the authors personal viewpoint on this specific area of investigation (potential benefits, drawbacks and vision of necessary future actions). Additionally, yet unpublished original data acquired by the authors is also provided in the manuscript to justify one of the points that is being discussed. From our experience, “review” articles tend to be more in-depth, as well as “stricter” and “dryer” in their form, restricting much of the possibilities for authors to show their personal attitude and, most importantly, disallow inclusion of the original experimental data (which we deem necessary in our case), whereas “perspective” format articles are intended to stimulate the discussion in the researcher community belonging to a given specific area of investigation, which is the main goal of this manuscript. We are also firm that this particular manuscript could be viewed as a rather superficial one, should it be considered a legitimate “review”, as it was initially intended and written as a “perspective” article for the reasons stated above and is not up to standards of the intentional “review” type manuscripts in our opinion.
We have done our best to address the typing, formatting and grammar mistakes in the second version of the manuscript. Some of the sentences throughout the text were simplified and/or divided as suggested by the Reviewer 2 (e.g. lines 50-65 are now two sentences and the comas were now replaced with semicolons for the sake of readability of internal punctuation in a “serial list”). However, none of the co-authors is the native English speaker and English language might still not be of the best quality. If, in case of acceptance, the external language improvements shall be deemed necessary by production team, we will proceed with the employment of a professional proof-reading service.
The authors would like to thank the Reviewer 2 for the review and drawing our attention to some of the issues of the original version of the manuscript. We have now submitted the updated version for Your consideration. Apart from a few typo corrections throughout the text, all of the relevant changes from the previous version are highlighted in yellow. Should the Reviewer 2 spot any other issues or consider that the past ones not adequately addressed during this round of revision, we are eager to tackle them in the next round of revision.
Reviewer 3 Report
The manuscript titled "Sorting out the superbugs: potential of Sortase A inhibitors among other antimicrobial strategies to tackle the problem of antibiotic resistance" by Nikita et al.,
The current initiative to write a general perspective on developing antivirulence therapy is a novel and generous idea. Although, i personally feel some more information should be added to this perspective...
- Figure 1, Although authors have discussed about antivirulence therapy in section 3.5 in brief, but it will be also good to present another picture with more detailed view of antivirulence therapy used currently (Please refer doi:10.1038/nchembio.2007.24).
- Also Phage therapy section and antimicrobial peptide sections can be sortened to give more focus on anti-virulence section. As the whole perspective is to give awareness about alternative approach rather than conventional approah.
- It would be great to include a section with challenges ahead for this alternative approach.
- Are these sortase A assisted protein anchoring is somehow connected with quorum sensing?
- It is a matter of discussion to conclude that anti-virulence therapy can not develop resistance with evolution (10.1038/nrmicro1818).
And at last, not being a native speaker, i too find it sometimes difficult to understand the phrasing of the sentence with multiple typo errors. It is advised to the authors to go through the manuscript very carefully before resubmission.
Author Response
Comments and Suggestions for Authors
The manuscript titled "Sorting out the superbugs: potential of Sortase A inhibitors among other antimicrobial strategies to tackle the problem of antibiotic resistance" by Nikita et al.,
The current initiative to write a general perspective on developing antivirulence therapy is a novel and generous idea. Although, i personally feel some more information should be added to this perspective...
Response: We are grateful for the Reviewer’s question and valuable suggestions on ways of improving this manuscript. We have now submitted the updated version for Your consideration. Apart from a few typo corrections throughout the text, all of the relevant changes from the previous version are highlighted in yellow. If our answers and corrections made to the manuscript do not please the respected Reviewer 3, we would be glad to receive further commentaries regarding the necessary improvements.
Figure 1, Although authors have discussed about antivirulence therapy in section 3.5 in brief, but it will be also good to present another picture with more detailed view of antivirulence therapy used currently (Please refer doi:10.1038/nchembio.2007.24).
Response: We thank Reviewer 3 for suggestion, however, given the intricacies involved in different antivirulence strategy subtypes, we believe that making another simple picture, which would show their conceptually different approaches would not do them justice they deserve, whereas a detailed picture would require an elaboration on the aforementioned intricacies, but this would be out of scope of this manuscript. We, nevertheless, see the Reviewer’s point and, following intense internal discussion on the subject of this commentary, have opted to make a concise elaboration on the concepts behind the antivirulence strategies that were mentioned in the first version of the manuscript. This solution also partially addresses the 2nd commentary.
The second (extended) version of the section 3.5. appears below:
The rationale behind antivirulence strategies lies in the assumption that “disarmed” pathogens can do no harm, thus the aim of such strategies is to interfere with the bacterial virulence factors that help the pathogen to either cause damage to the host or evade its immune system and persist within the host [50]. The current anti-virulence strategies tend to target the processes of bacterial quorum sensing systems, biofilm formation ability, as well as disassemble functional membrane domains and neutralize bacterial toxins via usage of small-molecule compounds that show corresponding inhibitory activity against at least one of the virulence factors of the pathogen of interest [51].
It is clearly evident that only a few cells of any pathogenic bacteria are of no immedi-ate concern to the host, however, strength of all pathogens “lies in numbers” that they eventually reach through inevitable propogation in appropriate environmental condi-tions. With growth of bacterial cell population also grows the neccessity for “communica-tion” between the cells, which is essential for coordination of whole community towards situations that are advantageous to population as a whole. This communication was re-vealed to be mediated by small signal molecules that autoinduce expression of particular genes [52]. The cell density-dependent process of such communication is termed “quorum sensing” [53] and there are indications that it might occur even between different species [54]. It is thought that as much as 4-10% of bacterial genome and ≥20% of proteome could be influenced by quorum sensing and the effects of quorum sensing range from harmless metabolic or phenotypic adaptations to modulation of pathogenicity related virulence (e.g. biofilm formation, toxin expression) [55]. Thus, interference with quorum sensing cas-cades that blocks bacterial virulence-associated “communiction” (“quorum quenching”) was outlined as a promising antivirulence strategy [56].
Whereas inhibition of the transcription might seem a more appropriate approach for some of the bacterial virulence factors, such as protein-based toxins, thus, eliminating the cause rather than a consequence, it is also possible to target these molecules after their synthesis. Studies in this direction have lead to the development of approaches that enable evasion from the destructive effects of toxins on the host through their neutralization by antibodies, toxin activity blocking by small molecule compounds [57], or even sequestra-tion of toxins in artifical liposomes [58].
Recently discovered bacterial functional membrane microdomains (FMMs), that re-semble lipid rafts of eukaryotic cells in both structure and function, were immediately proposed as yet another novel antivirulence strategy target due to the involvement of FMMs/FMM-associated proteins (e.g. flotillins) in biofilm formation, attachment, viru-lence, and signaling [59–61].
The ability of some pathogens to form biofilms, which are bacterial communities en-closed in structure formed by theirof extracellular matrix components, helps embedded bacterial communities to evade host defensive responses, mitigate environmental stresses and provides protection from antibiotics [62]. Pathogenic biofilm formation on the host tissues facilitates onset of a chronic bacterial disease that is considerably more difficult to treat than acute infections caused by pathogens in planctonic (“free”) state [63]. Thus, prevention of biofilm formation is also recognized as a prospective antivirulence strategy that strives to limit bacterial adhesion to surfaces or affect the extracellular matrix com-ponent production, and sometimes even destroy the extracellular matrix post factum, when biofilm has already been formed [51].
One of the particularly promising antivirulence strategies that is aimed at Gram-positive bacteria is the inhibition of bacterial cysteine protease – sortase (especially SrtA) activity by small molecule compounds [64].
Also Phage therapy section and antimicrobial peptide sections can be sortened to give more focus on anti-virulence section. As the whole perspective is to give awareness about alternative approach rather than conventional approah.
Response: We agree that some of the alternative strategies to combat antibiotic resistant bacteria (e.g. phage therapy, antimicrobial peptides) are, indeed, older than others, and have gathered greater research attention and body of evidence on their usage in therapeutic setting. However, we cannot agree that these approaches might yet be considered as conventional, because their employment in therapy globally is rather uncommon. Although, for example, phage therapy might be somewhat more accepted in some countries for historical reasons (e.g. polyvalent phage cocktails by "Микроген" (Microgen) are sold in pharmacies throughout the Russian Federation; personalized treatment in the form of patient pathogen strain specific phage cocktail formulation preparation options are available in Georgia’s Eliava institute and Poland’s Phage Therapy Unit of the Medical Centre of the Institute of Immunology and Experimental Therapy PAS), the global acceptance of such drugs is still non-existent, and their approval for human use in the Western countries, to our knowledge, is limited to individual cases on request of treating physicians in severe AMR pathogen cases. We are aware that there are numerous phage and antimicrobial peptide based formulations in different stages of clinical trials, and we are hoping that some of them will eventually prove their safety and efficacy to get officially recognized (registered) for human use and subsequently gain the widespread acceptance, which would be a great advancement for alternative strategies to combat antimicrobial resistance as a whole.
We would also like to mention that this particular “perspective” (Sorting out the superbugs: potential of Sortase A inhibitors among other antimicrobial strategies to tackle the problem of antibiotic resistance) was prepared for submission to the special issue "Solutions to Antimicrobial Resistance", which, to our understanding, covers a rather broad and diverse topic that, we believe, shall appeal to both, more general readership and investigators from specific topic-related areas, alike. We think that brief description of most prominent alternative antimicrobial strategies is necessary to introduce readers to different possible solutions in a broader context before narrowing down to a very specific and promising area of investigation (SrtA inhibition studies). Therefore combining Reviewer’s 3 suggestion #1 and #2 we have tried to retain these subsections in their entirety, but supplement them with additional information about the antivirulence strategies to provide greater focus on this subject, before delving into one of them (SrtA inhibition). If Reviwer 3 finds our solution and justification inappropriate, we can address these issues in the second round of revision.
It would be great to include a section with challenges ahead for this alternative approach.
Response: In our opinion, the future challenges for this approach are highlighted in the discussion and conclusion sections. We believe that the main challenge is related to standardization of the experimental procedures and result reporting practices of the SrtA research community. This would ensure the comparability of the further results acquired in the field and promote collaborations that could push more promising compounds into in vivo studies, thus speeding up their road to possible clinical studies. We are not sure if we should add a dedicated challenge section, as the challenges are already outlined in the seamless, and, hopefully, engaging way.
Are these sortase A assisted protein anchoring is somehow connected with quorum sensing?
Response: We believe that many of the antivirulence strategy targets are intertwined with each other. While we are not aware of the evidence that would reliably suggest that SrtA expression or activity is directly connected with quorum sensing, a somewhat “distant” link can possibly be established:
- Cell wall anchoring in Gram(+) bacteria occurs through a transpeptidation mechanism, which requires surface proteins with LPXTG motif-type sorting signal and SrtA enzyme. The genome of Staphylococcus aureus encodes >20 surface proteins with cell wall sorting signals. (Deletion of the SrtA gene or its inhibition abolishes the cell wall anchoring and surface display of SrtA substrates, thus interfering with the ability of cells to adhere).
- We know that bacterial adhesion to the host cells is a critical step for effective colonization of the host and onset of the disease; bacterial attachment to host cells also promotes biofilm formation.
- It is also known that bacterial behavior within biofilms is regulated by the phenomenon of quorum sensing, where bacteria release chemical signals and express virulence genes in a cell density-dependent manner.
Thus, in our opinion, the process SrtA-assisted protein anchoring can be indirectly connected with quorum sensing, although might not be dependent on it.
It is a matter of discussion to conclude that anti-virulence therapy can not develop resistance with evolution (10.1038/nrmicro1818).
Response: We think that any of the antivirulence strategies should undoubtedly exert weaker selection for resistance than conventional antibiotics as antibiotics affect cell viability in a direct way, whereas antivirulence strategies do not.
We did not state that anti-virulence strategies do not have potential to develop resistance in the course of evolution and we did not hypothesize on the overall development of resistance to anti-virulence therapy. In our manuscript we discussed various resistance emergence scenarios to SrtA inhibitory compounds and, in some cases, possible likelihood of their occurrence in comparison to other scenarios. Even more, we strongly believe that such resistance shall emerge, but it will come with the cost that shall be related to moderate decrease of pathogen virulence, as it is often in the case of antibiotic resistance mutation formation that are accompanied by the decrease in overall fitness (PMID: 25861385). To emphasize that these lines should be considered as a theoretical speculation, we have augmented the second sentence of the indicated paragraph with “We hypothesize that…”.
Moreover, there has been documented evidence of cross-resistance between antibiotics and some of the antivirulence approaches (PMID: 24014536).
And at last, not being a native speaker, i too find it sometimes difficult to understand the phrasing of the sentence with multiple typo errors. It is advised to the authors to go through the manuscript very carefully before resubmission.
Response: We have gone through the text once more; typing, formatting and grammar mistakes that were identified by us are now addressed. Some of the sentences throughout the text were simplified and/or divided as suggested (e.g. lines 50-65 are now two sentences and the comas were now replaced with semicolons for the sake of readability of internal punctuation in a “serial list”).
We apologize if language quality has hindered the readability of this manuscript. None of the co-authors is the native English speaker and English language might still not be of the best quality. If, in case of acceptance, the external language improvements shall be deemed necessary by production team, we will proceed with the employment of a professional proof-reading service.
Round 2
Reviewer 1 Report
I have read the authors' answers very carefully, however my views on some parts of the manuscript have unfortunately not changed.
As I have already said, from 3.1 to 3.4 they have no scientific interest for the purposes of the subject matter of the manuscript or add useful information to the discussion.
In the same way I believe that these manuscripts which contain experimental parts and typical parts of the reviews generate confusion in the reader. However, this is not the fault of the authors.
My observation derives from the specifications dictated by the "Instruction for authors" regarding the "Perspectives" manuscript type. I have reported below the part taken from the site
"Perspectives: Perspectives are opinion or commentary articles that express personal opinions about existing studies that have great impact on the antibiotics research community. Perspectives should have a main text of around 2000 words at minimum, with at least 20 references. The same requirements apply to communications, brief reports, and other similar publication types".
Author Response
I have read the authors' answers very carefully, however my views on some parts of the manuscript have unfortunately not changed.
As I have already said, from 3.1 to 3.4 they have no scientific interest for the purposes of the subject matter of the manuscript or add useful information to the discussion.
After the first round of revisions opposing opinions on this matter have been expressed even within the author team. Thus, on the one hand, we would be willing to remove these subsections from the manuscript, however, on the other hand, we are not fully convinced that this suggestion of Reviewer 1 would actually benefit the manuscript. During the first round of revisions we have explained the rationale behind our initial choice of the manuscript layout and the role of these subsections. Although we would like these subsections to be included in the final version of this manuscript, as it seems that 2 out of 3 reviewers are pleased with the current version of the manuscript that contains these subsections, we think that, at the current stage, decision on whether these subsections should be retained or excluded is dependent on the opinion of the academic editor/editorial team.
Should the academic editor/editorial team support the suggestion of Reviewer 1 to remove subsections 3.1-3.4, we shall comply with the request, but we also believe that in that case the change of the title of the manuscript from “Sorting out the superbugs: potential of Sortase A inhibitors among other antimicrobial strategies to tackle the problem of antibiotic resistance” to “Sorting out the superbugs: potential of Sortase A inhibitors to tackle the problem of antibiotic resistance” shall be required to better represent the contents of the revised manuscript.
In the same way I believe that these manuscripts which contain experimental parts and typical parts of the reviews generate confusion in the reader. However, this is not the fault of the authors.
My observation derives from the specifications dictated by the "Instruction for authors" regarding the "Perspectives" manuscript type. I have reported below the part taken from the site
"Perspectives: Perspectives are opinion or commentary articles that express personal opinions about existing studies that have great impact on the antibiotics research community. Perspectives should have a main text of around 2000 words at minimum, with at least 20 references. The same requirements apply to communications, brief reports, and other similar publication types".
The authors see the point of Reviewer 1 regarding the inclusion of experimental data in this article. However, we felt the need to give experimental evidence for one of the issues being discussed in the manuscript to back up our concerns. To our knowledge, this issue has not been explicitly raised in other published papers and seems to be largely overlooked in the SrtA inhibitory compound research field.
The choice of “perspective” type of manuscript was explained previously in an answer (excerpt from the answer given below) to commentary by Reviewer 2, who suggested that this manuscript is reminiscent of a review.
“We have found that the “perspective” format was the best fit for the contents of the manuscript, because after briefly giving readers some broader context and topic-specific evidence that is necessary to evaluate the importance of issues being raised and back up the points that are made, the manuscript largely involves presentation of the authors personal viewpoint on this specific area of investigation (potential benefits, drawbacks and vision of necessary future actions). Additionally, yet unpublished original data acquired by the authors is also provided in the manuscript to justify one of the points that is being discussed. From our experience, “review” articles tend to be more in-depth, as well as “stricter” and “dryer” in their form, restricting much of the possibilities for authors to show their personal attitude and, most importantly, disallow inclusion of the original experimental data (which we deem necessary in our case), whereas “perspective” format articles are intended to stimulate the discussion in the researcher community belonging to a given specific area of investigation, which is the main goal of this manuscript.”
Regarding the experimental part of this manuscript, we also see that the Reviewer 1 has stated in the checklist that the research design and methodology “can be improved”, but we do not see any suggestions on what exactly seems to still concern the Reviewer. During the first round of revisions the checklist stated that these things “must be improved”, we have made the changes we saw fit and the reply contained the following sentence: “Although we believe that the description allows replication of the results, if the Reviewer suggests that elaboration on something particular about the methodology is necessary, we would be happy to include this in the next round of revision.” No further suggestions on this subject were, however, received. We, the authors, believe that the provided description of methodology allows replication of the results acquired and the particular experimental design was planned well enough and is representative of the common practices in the field.
While we do not completely agree with some of the commentaries made by the Reviewer 1, we, nevertheless, are grateful and acknowledge the efforts partaken by the Reviewer in aiding us with the improvement of this manuscript and fixing some technical errors present in the initial version.
Reviewer 3 Report
The revised version of the manuscript has substantially improved and also the authors have answered all the questions raised. Hence, i dont have any further questions.
Author Response
The revised version of the manuscript has substantially improved and also the authors have answered all the questions raised. Hence, i dont have any further questions.
We are to thankful to Reviewer 3 for valuable suggestions that have helped to make this manuscript better.
We once again apologize if language quality has hindered the readability of this manuscript. None of the co-authors is the native English speaker and English language might still not be of the best quality. If, in case of acceptance, the external language improvements shall be deemed necessary by production team, we will proceed with the employment of a professional language editing service.